# A Review of Evolving Paradigms in Youth Studies

**Laura Guerrero Puerta** [1,2,3,4]

1 Department of Social Psychology and Education, University Pablo de Olavide, 41013 Seville, Spain; laura.guerrero.puerta@gmail.com
2 Department of Didactics, International University of La Rioja, 26006 Logroño, Spain
3 Department of Education and Social Psychology, University Isabel 1 de Castilla, 09003 Burgos, Spain
4 Reseach Group HUM-308, University of Granada, 52005 Granada, Spain

**Abstract:** This article focuses on the changes experienced by European youth because of the neoliberal globalised model. It analyses the impact of these socio-economic changes on school-to-work transitions and explores different theoretical perspectives (from the linear to pinball models) to understand them from a critical point of view centred on the individual. These transformations have caused the traditional markers of passage to adulthood to become diluted, generating de-standardised trajectories with possible "round-trip" states. The aim is to provide an understanding of youth-related phenomena under models described with the sociology of transition, exploring aspects such as reversibility and contextual influence on their life course.

**Keywords:** youth; neoliberalism–globalisation; school-to-work transitions; de-standardised trajectories; sociology of transition

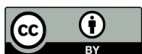

## 1. Introduction

Youth as an object of study has been an ongoing theme within the social sciences, anthropology, and education for over a century. This has caused a variety of paradigms to be developed that seek to define the category of youth. Upon examination of the trajectory of the production of the literature on this topic, it is evident that multiple and varied traditions, originating from various cultural and linguistic contexts, exist with scant interaction between them [1].

In the 1980s, in the wake of social transformations, studies began to appear that focused on the structural changes affecting youth. Giddens [2], with his analysis of post-Fordist societies, and Beck [3] and his theories on the society at risk, served as the basis for a new paradigm of youth studies. Consequently, a series of studies emerged in Europe, wherein biographies became a central element in an environment of increasing risk and transformation, where young people had to take responsibility for their own choices in the face of the weakening of welfare systems that had existed up to that point [4]. This gave rise to the biographical perspective, also known as the sociology of transition, which has remained a prominent feature of youth literature for the last thirty years [5].

The development of this biographical perspective, explains Sepúlveda [5], has been based on three fundamental pillars. The first pillar is the empirical study of the accelerated transformation of the labour market due to neoliberalism, which has facilitated gaining conscience about how this economic model has made the individual responsible for success or failure and has caused them to internalise a maximisation of individual productivity, leaving young people in a precarious situation. The second pillar is the acceptance of the theoretical proposals of authors of late modernity, who have included the notions of uncertainty, individual risk management, and reflexivity in the understanding of youth experiences in the face of the weakening of institutions and regulatory mechanisms of social life. The final pillar is the theoretical underpinning of the theoretical–

methodological paradigm of the life course, which has helped to overcome the strict, linear sequential interpretation of life experience that was prevalent in previous youth studies [6]. This paradigm was represented by Elder, who was a key contributor in situating the experiences of individuals in historical contexts and demonstrating that such experiences are dependent on the social, economic, and historical environment, wherein situated and interdependent relationships are essential factors in the experience of subjects throughout their lives [7]. This paradigm has been key to understanding the situated experiences of individuals across multiple disciplines (nursing studies and ageing studies, among others), providing a theoretical framework to work upon.

With these three pillars, the literature on youth from the perspective of the sociology of transition has approached the transition to adulthood of young people from different angles. This has caused the traditionally linear explanations of the transition to adulthood to give way to more intricate models that attempt to reflect the uncertainty with which young people move towards adulthood [8]. Authors such as Staff and Mortimer [9] summarise changes that have occurred and the main characteristics of this new youth experience, highlighting the following:

o    The individualisation of trajectories depends to a large extent on individual action and personal resources (understood as the specific capital accumulated by a subject). Processes such as starting and finishing studies, entering the labour market, or leaving the parental home correspond to less socially defined processes, and the weight falls more directly on individual decisions, with an increasing effect on the diversification of individual trajectories [9].

o    Related to the above, a second aspect that has to do with the individualisation of trajectories is the de-standardisation of social itineraries, whereby new living conditions have led to a loss of meaning of transition as a normative and linear process. The traditional model of dependency and preparation for adult life, which corresponds to the sequential transition from school or student life and entry into a profession, has given way to several additional transition variants with a higher degree of complexity and greater uncertainty about their temporal and spatial boundaries [9].

o    Finally, another characteristic of the times is that the youth transition has been temporarily prolonged, delaying the full independence of many young people and modifying the traditionally defined understanding of adulthood and adult roles. This has been explained, on the one hand, by the increase in the average number of years spent in education and the growth of the higher education market, and, on the other hand, by delays in starting a family or making decisions that require economic autonomy [9].

In this way, a consensus has been reached on the idea that the socio-economic landscape has been transformed dramatically, with profound implications for the way young people transition to adulthood. The shift from the Keynesian–Fordist capitalism of the mid-20th century to the neoliberal globalised capitalism of today has given rise to a new era of uncertainty, in which the responsibilities, milestones, and expectations associated with the transition to adulthood have become increasingly fluid. Therefore, it is essential to gain a better understanding of the broader context of the transition to adulthood to inform and shape effective social and educational policies for this age group.

This article will review the main theories that explain the changes in the transition to adulthood of young people that have occurred due to the neoliberal globalised model and the different theoretical approaches used to explain these changes. We will begin by exploring the socio-economic shifts in detail and outlining their implications for the transition to adulthood. Furthermore, considering the importance of metaphors in sociology for exploring and explaining social phenomena, as well as escaping conceptual traps and creating new social imaginaries, we will examine the research tradition on the transition to adulthood, the biographical perspective also known as the sociology of transition. We will

focus on different theoretical perspectives, such as the railway, yo-yo, and pinball models, to gain a critical and individual-centred understanding of the process of becoming an "adult", as well as their main limitations.

The goal of this article is to reflect on the changes that have occurred in the transition to adulthood of young people because of the neoliberal globalised model. By exploring the implications of these changes, we can gain a better understanding of the broader social context and its implications for young people. This, in turn, can help to inform social and educational policies that are more attuned to the needs of young people and better equip them to navigate the transition to adulthood. Such policies should consider the fluidity of expectations, responsibilities, and milestones associated with transitioning to adulthood and consider the various theoretical perspectives, metaphors, and limitations that have been discussed in this article. By doing so, we can create an environment that is more conducive to helping young people achieve the necessary skills for a successful transition to adulthood.

## 2. Navigating Uncertainty: Exploring the Implications of a Neoliberal Landscape on an Individual's Life Course

As previously mentioned, in recent years, there have been immense changes in the socio-economic paradigm that have had a considerable influence on the ambitions, self-conceptions, lifestyles, and plans of young people [5]. The Keynesian–Fordist framework, which had previously ensured work stability and was marked by the mechanisation of labour with highly repetitive and specialised tasks, has been abandoned. This has resulted in a greater distinction between manual and academic labour, and the state has reduced its involvement in regulating economic cycles and in developing welfare states through public policies [10,11].

These changes have given way to a flexible neoliberal model, which has drastically modified vital and social norms [12,13]. This model has been orientated towards global disorganised capitalism and can be considered a more radical neoliberal model [14]. In this way, neoliberalism has disrupted the perception of the future vs. present, blurring age-specific norms [15,16] and has appropriated, perverted, and economised the social and human sciences discourses that demanded greater social justice [14]. As a result, there has been an increase in precarious working conditions, a decrease in mobilisation and resistance to the growing inequalities, and a loss of rights achieved in past decades [17–19].

The neoliberal model has further modified the attribution of temporalities by transferring the responsibility of being an employable subject to the individual sphere, understanding power relations in a diffuse way and self-imposing new coercions in the name of freedom and self-realisation [18]. This has resulted in a need to interpret an individual's reality, give it meaning, and build narratives that unify their experiences [12,20–23]. It has also generated a "liquid" society, wherein the solid, which endures over time, has been altered in the face of the adaptive need of the market to give way to the changing and the fluid [12,23,24]. Moreover, it has caused a "corrosive" or "coercive" nature of these societies in which social security and solidarity have been eroded, and individualism, the internalisation of self-imposed coercion and inhibitions, has been deepened [12,13].

In this way, Beck [20] stresses that the main novelty of the historical stage in which we find ourselves is that individuals have appeared as the biographical solution to systemic contradictions, profoundly modifying the frameworks of action of individuals who have to move in a flexible and shifting panorama; thus, a whole system of "organised certainties", such as work, education, or family, has shifted towards "organised uncertainty", individualising social experience and diversifying the trajectories of individuals, who are now managers of their own lives [25,26]. This has had profound consequences on the transition to adulthood, whereby young people have been forced to take responsibility for this process, which has been significantly prolonged, and in which success or failure has been individualised, sometimes forgetting the importance that the social structure can



have both from the outset and in terms of the opportunities available in this transition [27].

In this way, neoliberalism has affected the development of youth, both in terms of their socio-economic standing and the ways in which they are perceived by society. By limiting the public spaces, resources, and institutions open to young people, neoliberalism seeks to neutralise their perceived threat to the status quo [28]. In the neoliberal era, the prescription for solving youth unemployment has shifted away from government intervention to teaching youth the "skills of employability". The "independent self" narrative which underpins neoliberalism is predicated upon colonial legacies, liberal ideologies, and capitalist logics [29]. This paints young people as incomplete humans on a journey towards a Western-style neoliberal adulthood, thus reinforcing exclusionary relations towards those who are unable to, struggle with, or simply choose not to follow this narrative. The neoliberal model has also caused the erosion of social security and solidarity, leading to a culture of individualism and self-imposed coercion. As a result, young people are increasingly responsible for their own successes and failures, and the importance of the social structures in which they live has been diminished [30,31].

To facilitate more ethical, socially just, and culturally appropriate care services, it is crucial to reassess how we both portray and relate to youth. At the same time, we must acknowledge that adulthood does not equate to independence and that being an adult involves periods of dependency that require cooperation and mutual caring as necessary to human life. This has been emphasised by international institutions such as the World Bank, which has suggested that "public policy needs to improve the climate for young people, with the support of their families, to invest in themselves" [30,31]. Consequently, the problems experienced by marginalised youth, such as poverty and social exclusion, are viewed as problems that can only be solved through market-based solutions [32]. This approach has resulted in the reinforcement of socially regressive policies, which are based on a deficit-based model of youth development [29].

The impact of neoliberalism on youth has had a significant effect on the ability of young people to access resources, opportunities, and institutions that are necessary for their development. To create more socially just and culturally appropriate services for young people, a joint approach is needed between governments, international institutions, youth organisations, employers, and civil society representatives. This approach should work to develop effective strategies for youth employment to build political consensus and policy coherence [33]. Only through such an approach can the negative effects of neoliberalism on youth be addressed.

## 3. Explanations of Youth Transitions to Adulthood: An Echo of the Changes That Have Taken Place around Youth

As previously stated, studying metaphors in sociology is important, as they can provide insight into how societies conceptualise themselves and the world around them. Metaphors can both enrich and reduce meaning and can be used to escape conceptual traps, create new social imaginaries, or explain complex phenomena [34]. The field of youth studies provides a particularly illustrative example of the importance of studying metaphors in sociology.

This section of the article will start with an exploration of the traditional concept of youth as linear, and then, we will explore three metaphors of youth transitions in more detail. The railway transition metaphor is based on a journey from dependence to independence and has been used in traditional youth studies. The yo-yo metaphor, which uses the idea of a pendulum swinging between dependence and independence, has been used to capture the more fractured and recursive nature of contemporary transitions. Finally, the pinball metaphor is a more recent addition to the literature that captures the sense of the young person as a pinball being bounced around the environment, with no clear point of arrival or completion. Each of these metaphors has implications for policy and practice and will be explored in turn. It is vital that we continue to explore the imaginative use of

metaphors in the field of youth studies to better understand the experiences of young people in our changing world [34].

### 3.1. Youth Explained as a Linear Transition

Linear transitions refer to a straightforward transition in which there are no major breaks or divergences. In Furlong, Cartmel, and Biggart's [35] view, linearity involves a smooth, one-way transition in which there are no major reversals. This model was understood as the norm in society during the industrial era up until the 1980s when youth was seen as a transitory period towards adulthood. It was characterised by a formative phase, followed by work, marriage, and the birth of children. Furlong, Cartmel, and Biggart [35] suggest that even during that period, a series of non-normative experiences of transitions to adulthood still existed, but government training schemes became an important mechanism for training and occupational socialisation, giving a response to those exceptions; as such, a transition could still be described as linear if the young person making the transition encountered short periods of unemployment or were "trained" in the context of a governmental programme and then "gained" a job.

In this way, youth was understood as a transitory and progressive period towards adulthood, exercising a "moratorium" role that accompanied childhood and served as preparation for the status of adulthood. In this context, life was organised through distinct stages, separated by certain rites of passage, which marked the end of one stage and the beginning of the next. Childhood youth was characterised as a formative phase. Adulthood, which began with leaving one's parent's home and getting married, was characterised by work. Finally, the last stage, old age, began and was characterised by withdrawal from the productive sphere [36].

This linear model of youth was developed based on the literature available in the fields of psychology studies. In psychology, the foundational work of Erikson and Inhelder and Piaget proposed theories based on stages of development [37]. Erikson developed Freud's psychosexual phases of development with adolescence and young adulthood as two distinct stages in an eight-stage theory of the life course; Piaget's developmental stages were anchored by the achievement of a stable adult identity, including the ability to think abstractly. Both pointed to adulthood as a point of arrival in this development [37].

The observation of youth life courses during the industrial era also played a significant role. The period of preparation necessary to ensure the proper use of new machinery to guarantee the productivity and competitiveness of a company increased significantly in comparison with the professional profiles of previous stages. This transition from youth to adulthood was explained based on a linear model (Figure 1), as it was made up of a series of irreversible and cumulative events that ultimately led to adult status. This model was characterised by a sequencing of age-based events such as education, employment/household work, marriage, and the birth of children. Men were seen as the workers and economic providers of households, and women were responsible for the care of the households and the offspring [27,36].

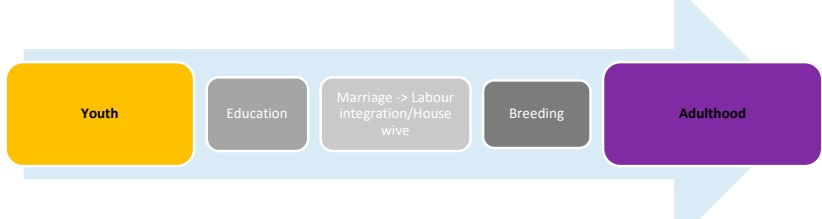

**Figure 1.** Linear transition. Source: own elaboration based on the study by Du Bois-Reymond and Blasco [36].

However, this linear model of youth transition has been challenged across the field of youth studies, with authors advocating a shift to "performative and processual identity" that understands the multiplicity and often circuitous nature of transitions to adulthood [37,38]. In this sense, Bob Coles [39], who was a pioneer in focusing his work on youth transitions, provided a more complex vision of these transitions, which he called "traditional". Coles made a critique of the conception of youth present in the literature up to that moment for placing excessive weight on the school–work transition, demanding a greater role for other types of transitions complementary to this one and seeking to understand youth experiences from a more concrete point of view.

In this way, the author explained that traditional transitions should be understood as the interplay of three highly interconnected transitions, namely, (1) the school–work transition, characterised by the shift from full-time study to work as the main occupation; (2) the domestic transition, which took place in the move from the family of origin to the family of destination, created by marital ties; and (3) the residential transition, which took place through emancipation from the family home [39]. This interrelation can be observed in Figure 2. The status achieved in any one of these transitions determined and was determined by the status of the remaining transitions. Thus, as he explained, a young person experiencing homelessness (the residential transition) would find his or her chances of starting a family (the domestic transition) or finding a job (the school-to-work transition) to be highly affected [40].

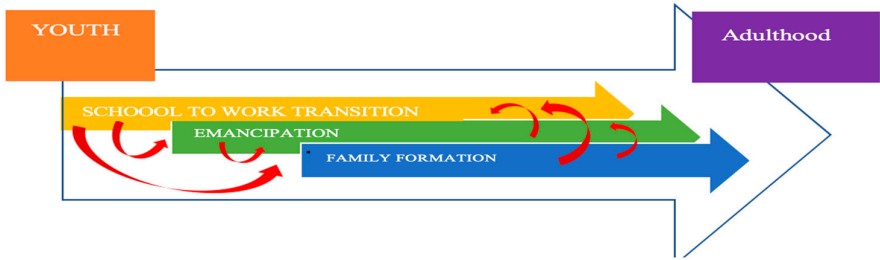

**Figure 2.** Traditional transitions according to Coles' [39] model. Source. own elaboration.

### 3.2. From Railways to Yo-Yo Transition to Explain Youth

This first attempt to study youth as an interrelated and complex whole was welcomed in the literature with great success and was followed by authors such as Furlong and Cartamel [26] and Wyn and Dwyer [41], who were quick to draw attention to the need to find a more adequate model to explain the youth experience, highlighting the inability of linear and traditional models to approach the new contexts and modes of expression of youth identity [42].

Since industrial times, production systems have undergone exponential change, requiring the workforce to have adaptive and renewable skills, which make the work experience more uncertain and changeable. Moreover, because of the extension of education, the educational stage has been lengthened in time, and similarly, a process of transformation has taken place in traditional family models, as well as in gender roles, and other areas of the life course understood as social institutions. This has led to the diversification of the routes to adulthood, giving rise to multiple ways of transiting youth [27,42].

As a result, the traditional markers of reaching adulthood have become more flexible and weakened and have acquired a reversible character, and age is no longer a predominant marker. Today, we can find people who are in training at an age above 30; young people who, after finding themselves unemployed or in undesirable working conditions, return to training; periods of emancipation that are reversed with a return to the home of origin for economic reasons; family break-ups; etc.

In this way, the concept of youth has changed significantly in contemporary society, with the emergence of a new form of adulthood. Young people today are characterised by

a greater degree of agency and autonomy, as well as a greater sense of identity exploration. This has been enabled by structural changes to the labour market, welfare systems, and family patterns, as well as by changes in individual choices. Furlong and Cartamel [26] explain these changes through comparative means of transport, which they use to explain how models of youth transition have moved from a "railway' model"—wherein journeys were determined as part of social reproduction and public railways were developed to aid the journey to adulthood—to a car model, wherein it is recognised that young people are at the wheels of their own cars, having a greater sense of control over the route and the importance of decisions and skills. However, the range and model of the car would still be important predictors of the outcomes of reaching the destination.

As a result, young people today are no longer bound by traditional markers of adulthood. Instead, they are characterised by a greater level of experimentation and a sense of being "in between". The emergence of this new form of adulthood has been termed "emerging adulthood" [43], "postponed generation" [44], or "new adulthood" [45].

The concept of emerging adulthood proposed by Arnett (2004) identifies certain features in the individual development process today, such as identity exploration, instability, focus on the self, a feeling of precariousness, and a greater emphasis on experimentation. This new concept of youth suggests that young people now have more opportunities to focus on themselves, explore different options, and cultivate individual aspects of their personalities without having to direct their biographies towards a definitive status.

The concept of postponed generation proposed by Mayer [44] reflects the growing trend of young people taking longer to transition into adulthood as it is made more difficult by the structural changes in the labour market and welfare systems.

Likewise, the concept of new adulthood proposed by Wyn and Woodman [45] suggests that young people today are having to manage a strong pressure to anticipate and quickly take control of their biographies to cope with the uncertainties of the present. This new way of being an adult is characterised by responsibility, choices between different options, a balance between spheres of life, the enhancement of extra-familial relationships, and a subjective maturation process that occurs in the absence of traditional status markers.

The changes in youth are strongly linked to the different structural changes in the labour market and welfare systems, as well as to the individual choices brought about by the "second demographic transition". This includes the increased participation of women in the job market, the diffusion of alternative models of families to marriage, and decreased reproduction. This has resulted in a new course of life for young women who are no longer univocally marked by reproductive tasks but instead are more individualised, following the increase in the levels of education and reduced dependence on the family [46].

Consequently, the linear concept of youth in the industrial stage no longer makes sense, as the routes to maturity have diversified, and there are multiple ways of experiencing youth [27]. This has led to the concept of youth as a sociological category being called into question, with authors pointing out that the concept of youth should now be understood as a relational category referring to the process in which age is socially constructed, institutionalised, and culturally legitimised based on a specific historical context [42].

Against this background, establishing the limits of youth has become a complicated task, which is the result of de-standardised and prolonged transitions in time, wherein dualisms such as adult/young person, single/married, and student/worker have lost meaning as they correspond to situations that must be considered under a wide range of intermediate and reversible, relatively transitory states. Today, we can find individuals with ages that do not correspond to the traditional conception of youth, but who, nevertheless, present markers associated with youth. These individuals hardly consider themselves young, but, at the same time, they find it difficult to situate themselves in adulthood [36]. In response to this, concepts such as that proposed by Whalter et al. [47], the Young-

Adult, have arisen, which attempt to take into consideration this intermediate stage, which can extend up to the age of 35, wherein an individual presents markers of both youth and adulthood, according to traditional models.

In their work, Biggart and Walther [48] argued for the use of the term "young adults" instead of "youth" and the metaphor of "yo-yo" transitions to describe the oscillations between adult and youth lives. This situation can be either involuntarily or voluntarily experienced, with economic, social, and personal resources often playing a role in determining the voluntariness of the transitions. For those with sufficient resources, this offers them the freedom to craft their own biographies, while those with scarce resources may face a more precarious route into adulthood. Additionally, these transitions may occur at different paces and in different areas such as education, work, lifestyle, family, sexuality, and civil life.

These changes in the conception of youth transitions can be seen in Figure 3.

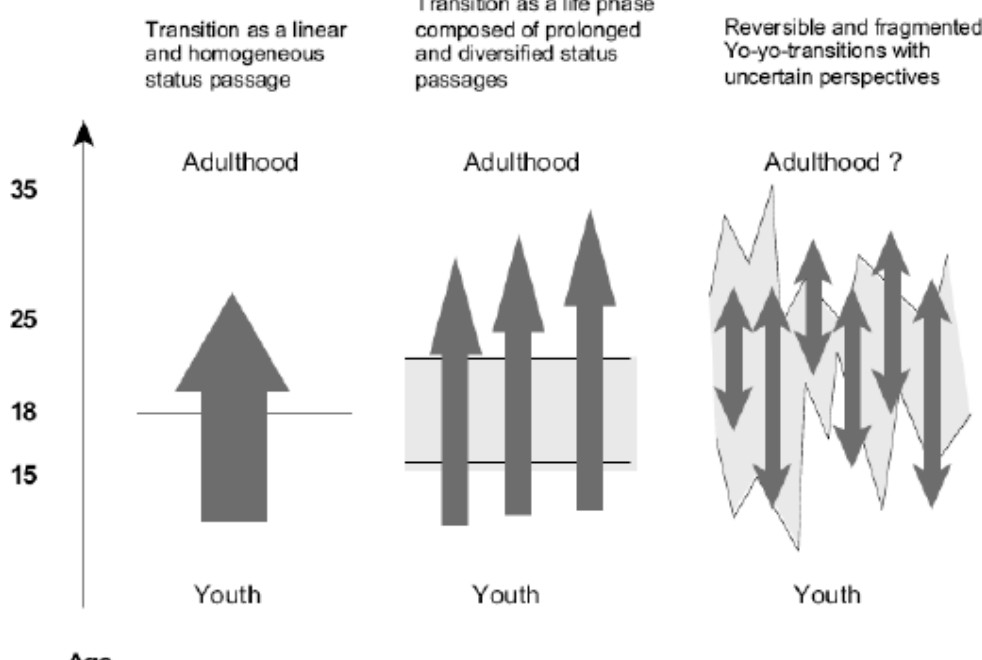

**Figure 3.** Youth transitions from linearity to yo-yo transitions. Source: Whalter et al. [49].

In consequence, the new transitional models described under the metaphor of "yo-yo" transitions, allegorising the toy that goes up and down by rolling up on itself, have been described based on multiple possibilities around the transition to adulthood [36,50] including the following:

- Young adults who are forced to alternate between precarious work, unemployment, and compensatory training and who have limited resources.
- High-income young adults who make use of their freedom to choose possible options according to their needs and preferences.
- Young adults who want to improve their situation by combining training and employment but who are forced to give up their professional and training desires in favour of standardised and limited trajectories.
- Young people who prolong their dependence on their parents in the face of insufficient social coverage in the face of youth unemployment.

This range of possibilities in the yo-yo transition highlights the debate surrounding youth, demonstrating that the changes that have occurred in the institutions of modernity

have caused the mutation of social and individual temporalities. This is largely due to the crisis of social states and particularly the changes in their main integration axis: work. While in previous times, life events were standardised and institutionalised, with access to the labour market and work activity serving as the backbone, today we must also consider other fragmented transitions instead of the traditional transition from school to employment. These transitions move at different paces and follow varied logics, which can be synchronised and even reversed [27].

In this way, the new models of youth transitions can be better understood if we complement them with the concept developed by Alheit [51] of biographisation, which some authors [36] consider to be the other side of the coin of yo-yo transitions. This term seeks to draw attention to the internal processes of opportunity and risk management and to how the individual makes sense of his or her own biography in this relationship, in a context in which the "normal biography" is defined solely through gender and class roles has disappeared. However, it also reviews the constant task of reflection that an individual must carry out on his or her own decisions, such that self-evaluation has become a recurrent and necessary task. Thus, this biographisation and the evolution of youth transitions, both of which are outcomes of the modernisation process, are highly dependent on the existing structures in the contexts in which they take place, and their development and outcomes will depend on the country in which they occur [36].

*3.3. The Pinball Metaphor of Youth Transitions: Exploring the Precariousness of Mobility and Labour Markets*

The most recent attempts to explain youth transitions have sought to capture the uncertainty experienced by young people in the construction of their biographies, including the effects of mobility (geographical or otherwise) encouraged by the institutional sphere. Cuzzocrea [52] introduces the metaphor of pinball to capture these intensifying patterns of mobility and precarity in youth labour markets.

This metaphor is based on the classic pinball game, in which a small ball is propelled with a plunger, and the objective is to score points by causing the ball to interact with the game system, preventing it from rolling backwards and leaving the board. By using the term pinball to refer to youth transitions, Cuzzocrea [52] creates a metaphor to highlight the possibility that, despite participating in the game, there may be a lack of agency in many young people with respect to controlling their career trajectory [53]. The mechanisms may be available, but, as with a pinball machine, the direction the ball takes at any given moment is unpredictable, as is the effectiveness of the moves made to keep it in play [52].

Furthermore, the pinball game has a fixed duration, but with repetitive episodes of movement, whereby each push of the ball on the table returns the individual to the starting point, instead of contributing to a cumulative trajectory. This metaphor can be related to the idea of the "epistemological fallacy", referring to that which creates the illusion of equality while hiding the persistence of inequality [26]. This is sometimes caused by a policy focus on employability and activation policies [52].

The individualisation of the experience of playing pinball is also an important consideration and is, according to the author, its most obscure interpretation. Connection with others takes place primarily in terms of the amount of success one generates in relation to others. Similarly, a career can become competitive within a cohort. A cohort may share similar goals, but the "game" is still based on individual resources, whereby players face the high competitiveness characteristic of neoliberal contexts. Thus, constant effort is required just to "stay in the game" of life and invest in one's own skill set, although the skills acquired may not even be aligned with the skills needed and may end up not being useful [52].

At the same time, the pinball metaphor leaves room to appreciate that attempts to move from education to the labour market can have a playful and personal developmental dimension. Just as the pinball game has goals to meet and obstacles to avoid, so too do

young people. The "pinball" metaphor, therefore, expresses a self-interested—and externally exploited—strategy for becoming an adult, and one that we must recognise as such. In this sense, Cuzzocrea [52] points out that this metaphor also allows us to introduce an additional temporal and spatial dimension to young people's transitions, recognising that certain stages involve taking risks with uncertain and unpredictable outcomes, which may materialise as an episode of mobility.

The idea of speed inherent in a pinball game is also key in the context of "social acceleration", whereby technological, economic, social, and cultural processes are in continuous change [54] and in which these early career workers are immersed. This facet allows us to visualise elements of space and time in a complex and continuous transition [34].

The pinball metaphor of youth transitions is a useful tool for understanding the current state of youth labour markets. It highlights the lack of agency and unpredictability that many young people are facing, as well as the individualisation of the experience. Moreover, it brings to light the idea of social acceleration and reveals the complex and continuous transition that young people must go through to succeed in the labour market. Finally, it recognises the effort that young people need to invest to stay in the game and the fact that the skills acquired may not even be aligned with the skills needed to reach their goals.

By introducing this metaphor of youth transitions, Cuzzocrea [52] seeks to bring awareness to the effects of the precariousness of employment and mobility in the lives of young people. The pinball metaphor is thus a useful way to better understand the current state of youth labour markets and to highlight the challenges that many young people face to achieve their goals, which, sometimes, are accepted in the hope of better outcomes. Moreover, it can bring attention to the temporal and spatial dimensions of youth transitions as a complex and continuous process.

## 4. The Importance of a Comparative Perspective for a Better Understanding of Youth Biographies

Despite the general trends of the de-standardisation, individualisation, biographisation, and prolongation of trajectories and the metaphors that have been explored so far, there are considerable variations in youth transitions depending on the institutional contexts of each country [55–60]. Many authors [55,60–64] have drawn attention to the importance of the social welfare systems in different states as a framework for the development of youth transitions, insofar as they serve as the structural basis on which young people find opportunities, negotiate their expectations, and make decisions. Based on this idea, several attempts have been made to explain the transition regimes that take place in the different countries of the European Union [55,61,62].

This concept of transition regimes refers to national configurations that regulate the life course and distinguishes between socio-economic, institutional, and cultural clusters and their interactions with agency. With the notion of a regime, they refer to the fact that the regulation of trajectories expands beyond institutional governance to include individual biographical constructions. Thus, by using transition regimes, it is possible to distinguish the various configurations of power and normality that are present in different states and how they respond to and organise the binomial of social inclusion and exclusion [65]. The different models of transition regimes have been consistent in showing these institutional-context-dependent variations; however, Walther's [61] model stands out for its enquiry into cultural characteristics as key elements for a better understanding of regional variations in transitions in addition to the parameters studied in the other models, which focus solely on the economy and social policies [60].

In this way, the model proposed by Walther [61] focuses on biographical choices that are contextualised in specific cultural and institutional structures to explain the behaviour of the youth population in relation to employment and the constructions of the couple and the family in different European countries [64,66]. On this basis, it proposes four

transition regimes, under which the main countries in the European continent are grouped according to their characteristics. These are described below:

- The universalistic transitional regime: Exemplified by Sweden, Denmark, and Finland, this regime is characterised by an integrated system of education that combines vocational training and university education, which can be adapted to individual trajectories. In the labour market, employment policies focus on job security and motivation and are accompanied by a range of family and gender policies that promote high levels of gender equality [61].

- The employment-centred transition regime: This regime is characterised by a selective school system in continental countries, such as Germany, France, and the Netherlands, that allocates younger generations into occupational careers and social positions. Vocational training is a central part of this system and is relatively standardised, with school-based, company-based, and mixed approaches in France, Germany, and the Netherlands, respectively. Labour markets are divided into a protected core and precarious peripheries, with women being underrepresented in the core. Social security systems differ, with social insurance providing higher levels of compensation for those in standard work arrangements and a residual social assistance system for those in the peripheries. Recent workfare elements have challenged these systems, with young people not automatically entitled to benefits if they have not paid enough social insurance contributions. Programmes aim to compensate for learning or social deficits, rather than providing access to regular training and employment, except in the French "emploi-jeunes" programme [61].

- The liberal transition regime: Specific to the United Kingdom and Ireland, this regime emphasises individual responsibility for one's own welfare, including rapid and stable integration into the labour market. The UK and the Republic of Ireland have a system of governance that places emphasis on individual rights and responsibilities rather than collective provisions. Education up to the age of 16 is largely comprehensive in the UK, while differentiated routes exist in Ireland. Post-compulsory stages have been developed and diversified to provide a range of vocational and academic options, with various entry and exit points. This is seen as an investment to prepare individuals, who are regarded as the "entrepreneurs" of their own labour, for self-sufficiency. Benefits are now linked to citizenship status and are generally low, limited in duration, and dependent on actively seeking employment. There is still a belief that youth is a transitional phase that should be replaced as soon as possible with economic independence, leading to the view that youth unemployment or disadvantage is due to a culture of dependency. Labour market flexibility is high, but the level of qualification in the workforce is low, resulting in a high rate of female employment. This has, however, also led to precarious conditions for many, including the need to provide childcare, which is mostly arranged through the private sector and in Ireland is reinforced by a strong Catholic family ethic [61].

- Southern European countries or the sub-protective regime: The sub-protective transition regime mainly applies to southern European countries, such as Italy, Spain, and Portugal. These countries have a low percentage of regular work arrangements and a high rate of unprotected living conditions. School is structured until the end of compulsory education, but there is still a high rate of early school leaving, with child labour being especially prevalent in Portugal. Vocational training is not well developed and is mainly provided by professional schools, with limited involvement from companies. Young people are not eligible for social benefits, so they are often stuck in precarious jobs, either in the informal economy or on fixed-term contracts. Labour market segmentation and a lack of training lead to high rates of youth unemployment, particularly affecting young women who also face difficulties due to the lack of public childcare facilities. Higher education is important for providing status in the waiting phase, but many students drop out before finishing their degrees or become over-qualified. The main policy objectives are to prolong school participation,

integrate and standardise vocational training, and create jobs, including incentives for employers, career guidance, and assistance in self-employment. The overall goal is to provide youth with a regularly institutionalised status, be it in education, training, or employment [61].

This typology is effective for distinguishing clear differentiating features between the different states wherein young people transition to adulthood. However, the model at hand also has clear limitations, as it does not include Central and Eastern European societies and other non-Western contexts, which makes it a heuristic rather than a descriptive approach, open to diverse contexts and different normalities that by their changing nature need to be subject to periodic revision [61].

## 5. The Duality of Agency and Structure: A Common Basis in Biographical Studies

When exploring the different metaphors used to explain youth transitions, an inherent issue emerges regarding the interrelationship between individual agency and the social structure of young people's life courses [67]. Mayer [68] contends that these new interpretive models are based on a combination of a structural theoretical perspective with the theory of individual action, i.e., agency and biographical choices, and their interrelation.

Emirbayer and Mische [69] put forth the concept of agency as a dynamic, temporal process of social engagement that is driven by past experiences but also directed by the capacity to project future possibilities and evaluate the present moment. This process involves considering the iterative, habitual elements of the past, the projective elements of the future, and the practical-evaluative elements of the present. In these ways, agency is understood as the capacity of individuals to move towards a goal and make choices, which are determined by the institutional framework and socially constructed conditioning factors. The structure, meanwhile, is a material, normative, and social framework, which has the capacity to condition the development options of agency, offering different viable alternatives to the action of individuals. This creates a binomial, in which agency manifests as a choice, and choice is only possible if there are structural alternatives available [42].

Such changes in life experience, such as the formation of a couple, leaving the family home, educational transitions, or entering the labour market, should be read as part of broader trajectories that are determined and give meaning to the life course as part of the historical moment and place in which they take place [7].

Therefore, and considering the previously explored transition regimes, youth is shaped as a stage that is characterised by a series of highly complex processes, in which structural and subjective conditions interact in a specific way according to the society in which a young person's life course takes place and the social condition of the individual, recognising that young people are not exclusively determined by micro- and macro-social factors but are instead active agents managing their own lives [1].

To explain this agency–structure tension, a series of metaphors have been developed in the literature to reinforce the processes experienced by young people. Metaphors such as "niches" or "routes"—which were used in the 1960s/70s to define the youth experience and refer to the structural frameworks that defined the youth condition configured as a linear experience—have been replaced with concepts such as "navigation", which place greater emphasis on the duality of agency and structure that is present in the transitions characterised as a yo-yo or pinball [5]. Furlong, one of the leading exponents of the analysis of youth from a biographical point of view, explains this by arguing that the transformations that took place during the 1980s and onwards have encouraged the concepts with a structural analytical basis to lose validity, weakening and intermingling with cultural studies [42].

The implications of viewing youth transitions through the agency–structure binomial are significant. Firstly, it gives greater importance to individual agency, allowing the

recognition of biographical choices and paths within the life course of young people. Secondly, it offers a wider range of possibilities to analyse the different levels of complexity experienced by young people, from micro-level decisions to macro-level social structures in neoliberal times. Finally, understanding the tension between agency and structure allows us to analyse the implications of different historical moments in the lives of young people and to develop ways to support them. In conclusion, understanding the interaction between agency and structure is essential for youth studies as it provides insight into the complex transitions experienced by young people.

## 6. Criticisms of the Sociology of Transition

When discussing the unintended consequences of understanding and portraying youth as a transition to adulthood, it is important to understand the implications of the standard model of adulthood. Youth has been seen as a transition to adulthood, with five markers of adulthood (such as completing education, financial independence, marriage, and parenthood) as the only legitimate model for successful transitions. However, there has been a lack of recognition of young people's multiple ways of navigating transitions and multiple ways of being adults, as well as the relational nature of adulthood and the social value and functions of youth "immaturity". This has led to a cognitive hierarchy between youth and adulthood as stages in the life course and a narrative of competition whereby young people are stigmatised as immature or lazy if they are unable to meet the prescribed goals. To move forward, it is necessary to re-adjust contemporary ideas of adulthood and to build a more "updated" and inclusive idea of transitions to adulthood and of adulthood itself [43].

The current model of adulthood, which is based on the experience of white middle-class young men in the Global North, has been generalised and pushed to fit in women, minorities, youth belonging to low-income families, and young people in non-industrialised countries. As a result, many young people are struggling to achieve the traditional markers of adulthood, with unemployment, economic instability, and in-work poverty becoming common experiences. Furthermore, the standard model of adulthood fails to acknowledge young people's multiple ways of navigating transitions and multiple ways of being adults, as well as the different life conditions from which different young people start their paths of transition, making it impossible for some to achieve the standard [43].

One of the criticisms that most radically questions the sociology of transition comes from feminist theory. These theorists, following the gender perspective, question the convenience of developing a model of approaching youth which, according to them, places paid employment as a central element of youth transitions. This is especially relevant given the characteristics of contemporary neoliberal societies, in which the burden of care and housework are not recognised in economic terms. In this way, this point of view of youth as a transition may be problematising a group of young people that does not necessarily have to be in conditions of vulnerability, or at least not for the reasons given, while at the same time devaluing these activities linked to care [1,70].

In addition, there are many authors who criticise the structure–agency dichotomy on which these studies are based as these debates constrain youth studies to an ontological point of view, insofar as agency is a vague and ill-defined concept and therefore difficult to analyse, specifically in times such as the present, wherein neoliberalism has interfered in power relations, causing a self-imposition of new coercions that are accepted and naturalised by the people living in this era, and, therefore, the agency of young people has been infused by certain neoliberal ideas [18,71]. In other words, neoliberalism's pervasive influence has led to the acceptance and internalisation of certain ideas and norms, which shape and limit the agency of young people in contemporary society.

In this sense, Threadgold et al. [71] point out that the new economic models, characterised by the emergence of the immaterial, require new norms in which a new direction is given to the way in which we understand subjectivity, which must be sustained and rethought based on how the sociological and political sphere have approached the period

of youth and what relationships have been associated with this concept. Especially in a scenario where immaterial work has taken on an unprecedented prominence, becoming the engine of growth in knowledge societies, the product of work has also become subjective, insofar as work creates a product that, precisely because of its intangibility, must be connected and socially repositioned to acquire value, problematising the individual perception of productivity. This is supported by authors such as Sepúlveda [5] who assert that the lack of consideration of the conditioning produced by social structures, and the way in which these structures are recreated and interpreted by the actors, represents one of the fundamental limitations to the way in which the search for an understanding of youth processes is approached.

This critic possesses the idea that to move forward, it is necessary to rethink our understanding of youth transitions to adulthood and to build a more "updated" and inclusive idea of transitions to adulthood and of adulthood itself. This would involve rethinking the metaphor of youth as a transition to adulthood and acknowledging the social value and social functions of youth. It would also involve giving visibility to the alternative ways of navigating transitions elaborated by contemporary young people and recognising that new perspectives on adulthood might emerge from practices and spaces that are not normally considered in actual narratives about transitions to adulthood. Finally, it would involve elaborating new ways of assessing the journey to adulthood, eliminating the correlation between traditional markers of adulthood and maturity and drawing on the practices of marginal segments of the young population, recognising that reaching adulthood is not about rites of passage, but it has more to do with the capacity to care for others and the world around us [43].

In response to these criticisms, authors such as Furlong [72] and Cuzzocrea [52] have argued that the current metaphors present in youth studies should be seen not as dogma but rather as an open invitation to rethink youth, going beyond the traditional model based on linearity and biological development. Moreover, they defend that a metaphor does not crystallise a phenomenon but is intended to enhance the visualisation of an idea of a conceptual nature, which, in this case, is characterised by being highly dynamic and not sufficiently studied or conceptualised. Following this argument, the biographical literature can be used as a starting point for creating new metaphors that allow a better understanding of the conditions under which young people live today.

## 7. Conclusions

This article has extensively analysed the deep-seated transformations that have come to fruition in recent years, triggered by changes in the social realm and the labour market, and intensified by neoliberal policies. The result of these neoliberal alterations has been a shift in the responsibility of navigating the transition to adulthood, which now lies upon the shoulders of young people. Additionally, the traditional markers of passage to adulthood have become increasingly blurred, giving way to de-standardised trajectories, wherein a "back and forth" pattern can be observed.

The metaphors and models portrayed in this article are not without flaws, and there are certain aspects that need further attention and research within the youth literature. Nevertheless, these models prove to be highly enlightening when it comes to interpreting phenomena associated with young people, such as school dropouts and NEETS. To fully understand the current situation and its implications, it is important to investigate the reversibility of certain processes and consider the influence of both local and supranational contexts on young people's transitions. Furthermore, it is essential to assess how neoliberal policies have impacted the lives of young Europeans, and how these changes are perceived by them.

The concept of youth as a transition to adulthood is complex and requires the recognition of alternative strategies and practices for reaching adulthood that have been developed by young people, such as environmental commitment and better mental health awareness, among others. Thus, it is important to construct a more updated and inclusive

notion of adulthood and transitions to adulthood that takes into consideration the right to be young and experiment, fail and enjoy, as well as the capacity of marginal positions to offer a unique point of view and potential, while, at the same time, seeking for greater social cohesion and loosening the responsibility placed on individuals by neoliberal societies. By understanding youth transitions in this new way, a more equitable and just society can be achieved.

In conclusion, this article has explored the deep transformations that have taken place in recent years, and how they have shifted the responsibility of navigating the transition to adulthood onto young people. It has highlighted the need to explore the influence of contexts on young people's transitions and to recognise alternative strategies and practices of maturity developed by young people. Ultimately, this article calls for a rethinking of the traditional idea of adulthood to construct a more equitable and just society.

**Funding:** This research is associated with the project funded by H2020, YOUNG_ADULLT. But funding has being fully assumed by the authors. The APC was funded by MDPI through reviewer vouchers, University of Granada IOAP program, and partially paid by author.

**Institutional Review Board Statement:** Not applicable.

**Informed Consent Statement:** Not applicable.

**Data Availability Statement:** Not applicable.

**Conflicts of Interest:** The authors declare no conflict of interest.

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
