# Peer review of "A Review of Evolving Paradigms in Youth Studies"

_societies, doi:10.3390/soc13060136_

Round 1

Reviewer 1 Report

see attachment 

I'm not a mother tongue, but I have the impression the english version is a direct translation from your mother tongue; often the sentences are too long and this make them sometimes unclear. I would strongly suggest a mother tongue revision,

Author Response

Dear Reviewer,   Thank you for taking the time to review our paper. We have taken your feedback into consideration and have done our best to address your comments and suggestions.   The title and type of paper has been changed to a literature review. Additionally, the concepts of “postponed entrance to adulthood”, “young adulthood”, and “emerging adulthood” have been added to Section 5. The idea that youth studies need a new approach has been included in Section 6. The references have been modified and some new ones have been added to improve the lack of precision. A complete restructuring of the paper has been made, as well as a re-write and proofread in English. The conclusions have been rewritten to reflect our point of view more clearly and to explain the added value of our paper to the scientific debate.   We hope this addresses your concerns and look forward to hearing your thoughts.   Sincerely, Author

Reviewer 2 Report

The article aims to reflect on the changes experienced by young people in the transition to adulthood due to the neoliberal-globalised model. The text does not provide novel knowledge or evidence on the widely internationally debated topics. Despite this, the contribution has the merit of approaching the complexity of transitional models, countering widespread neoliberal narratives focused on workfare activation and individual responsibility and providing a more comprehensive understanding of youth integration and school-to-work transitions. The author shows a good knowledge on the issues at stake and the literature quoted is appropriate. However, I consider that as the impact of socio-economic changes on school-to-work transitions is a key issue in the paper, it is not covered in depth in the current version of the contribution. I suggest minor changes to this section before publication.

Author Response

Dear Reviewer,   Thank you for taking the time to review our article. We appreciate your feedback and suggestions for improving the paper. We have taken your comments into consideration, and have made the suggested changes to the text. We have revised the section 2 and 3 adding information on socio-economic changes and their impact on school-to-work transitions, and analysis to this section. We hope that these changes meet your expectations, and that our paper can now be accepted for publication. Thank you again for your time and consideration. Sincerely, Author

Round 2

Reviewer 1 Report

Soundness and consistency of the article have been really improved, especially in the first part of the text. I'd like however making some minor remarks:

99 I would substitute linear with railway (vs. car model) instead of linear, as yo-yo and pinball are metaphors (I will insert her the concept of methaphors as at 195 you write "as already said)

202 the linear is not a metaphor! the corresponding metaphor of this transition is the railway. Therefore also the title should mention railway and not linear, as you then insert Pin-Ball explanation

556-565 you present another methaphor raylway and car model, Why don't you present it in the previous metaphors section?

501-505 How can German, French and Dutch edu systems be inclusive and at the same time selective and standardized? 

514 Have Southern EU countries a rigid edu system? Formally they are more permeabel of the above mentioned and moreover at 530 you write "by a non selective education system..."

632-635 it is not clear to me what you mean, perhaps something is missing

648 What does it mean "rediscovering" the metaphor of youth as a transition

In general pay attention to repetitions in text (for example 454-456 are very similar to 433-435; 226-229 is similar to 217-219: skip? integrate? substitute; 613-616 already said before) and to the use of They-their-them-that, sometimes it is not clear to what are referred ( for example: "that" in 462; "they" 478 and so on.)

684 you equate maturity (what the source of this concept?) to environmental committment, better mental health awareness and co-housing, however are we sure that adulthood implies this? I think I understand what you mean, but perhaps this sentence need a better formulation. 

Author Response

Dear Reviewer,   Thank you for your feedback and comments on my article. I have taken your remarks into consideration and have made the necessary changes.   In response to your comment about substituting linear with railway (vs. car model) instead of linear, as yo-yo and pinball are metaphors, I have done so.   In response to your comment about the linear not being a metaphor and the title needing to mention railway instead of linear, I have reformulated the section.   In response to your comment about presenting the railway and car model metaphor in the previous metaphors section, I have done so.   In response to your comment about how German, French and Dutch edu systems can be inclusive and at the same time selective and standardized, I have rewritten the section with a new reading of Whalters transitions regimes.   In response to your comment about Southern EU countries having a rigid edu system, I have rewritten the section with a new reading of Whalters transitions regimes.   In response to your comment about it not being clear what I meant in 632-635, I have reformulated the section.   In response to your comment about rediscovering the metaphor of youth as a transition, I have revised and skipped this section.   In response to your comment about maturity being equated to environmental commitment, better mental health awareness and co-housing, I have reformulated this paragraph.   In response to your comment about repetitions in the text and the use of they-their-them-that, I have revised and skipped this section.   All information that is new has been highligted in green, and significant changes has been marked with yellow.   Once again, thank you for your feedback and comments, as they have been of invaluable significant for improving my paper.   Sincerely, Author